# The Characterization and Beta-Lactam Resistance of Staphylococcal Community Recovered from Raw Bovine Milk

**DOI:** 10.3390/antibiotics12030556

**Published:** 2023-03-10

**Authors:** Nisa Sipahi, Ertugrul Kaya, Cansu Çelik, Orhan Pınar

**Affiliations:** 1Traditional and Complementary Medicine Applied and Research Center, Düzce University, 81620 Düzce, Türkiye; 2Medical Pharmacology Department, Medicine Faculty, Düzce University, 81620 Düzce, Türkiye; drekaya@yahoo.com; 3Food Technology Program, Food Processing Department, Vocational School of Veterinary Medicine, Istanbul University-Cerrahpasa, 34320 Istanbul, Türkiye; cansu.celik@iuc.edu.tr; 4Equine and Equine Training Program, Vocational School of Veterinary Medicine, Istanbul University-Cerrahpasa, 34320 Istanbul, Türkiye; orhan.pinar@iuc.edu.tr

**Keywords:** staphylococci, biofilm, beta-lactamase, coagulase, virulence, *mecA*, *blaZ*

## Abstract

Staphylococci is an opportunistic bacterial population that is permanent in the normal flora of milk and poses a serious threat to animal and human health with some virulence factors and antibiotic-resistance genes. This study was aimed at identifying staphylococcal species isolated from raw milk and to determine hemolysis, biofilm, coagulase activities, and beta-lactam resistance. The raw milk samples were collected from the Düzce (Türkiye) region, and the study data represent a first for this region. The characterization of the bacteria was performed with MALDI-TOF MS and 16S rRNA sequence analysis. The presence of *coa, icaB, blaZ*, and *mecA* was investigated with PCR. A nitrocefin chromogenic assay was used for beta-lactamase screening. In this context, 84 staphylococci were isolated from 10 different species, and the dominant species was determined as *S. aureus* (32.14%). Although 32.14% of all staphylococci were positive for beta hemolysis, the *icaB* gene was found in 57.14%, *coa* in 46.42%, *mecA* in 15.47%, and *blaZ* in 8.33%. As a result, *Staphylococcus* spp. strains that were isolated from raw milk in this study contained some virulence factors at a high level, but also contained a relatively low level of beta-lactam resistance genes. However, considering the animal–environment–human interaction, it is considered that the current situation must be monitored constantly in terms of resistance concerns. It must not be forgotten that the development of resistance is in constant change among bacteria.

## 1. Introduction

The presence of pathogens in foods of animal origin is a common healthcare issue. Some virulence factors of pathogens and resistance to antimicrobial agents constitute the basis of this problem. Also, the fact that the antimicrobial resistance characteristics of these pathogens, which are mostly opportunistic, are constantly changing, makes it difficult to monitor and solve the problem [1,2].

Although the microbial load of milk is low when it is first obtained, the variety of microorganisms it contains is important because it may contain mastitis factors that affect animal health and the milk quality. Mastitis is an inflammation of the mammary gland and the majority of infections are caused by *Staphylococcus* spp., *Streptococcus* spp., and Gram-negative bacteria [3,4]. Some of these factors are contagious and some are environmental. Knowing the microbial load obtained from raw milk and identifying microorganisms are important in terms of determining the subchronic potential. Despite many studies conducted on mastitis in Türkiye and around the world, it is still common in dairy cattle breeding. Therefore, antibiotics are used for its treatment [5,6].

The most common isolated bacterium from milk is *Staphylococcus aureus*. *S. aureus*, which is persistent in the normal flora of milk, sometimes causes toxic syndrome and staphylococcal food poisoning in animals and humans. Previous studies reported coagulase-positive (CoP) and coagulase-negative (CoN) *S. aureus* in milk and dairy products [7,8]. Coagulase characterization is considered a virulence factor for *Staphylococcus* spp. [9]. *Staphylococcus* spp., whose other virulence characteristic is biofilm formation, can sustain its existence in milk in a more stable way by forming a biofilm [3].

Another important issue is the microbiome aside from the opportunistic pathogen content in milk. Resistance genes in the microbiome pose a risk to animal and human health. Antibiotic resistance is already a growing crisis. This reduces the effectiveness of drugs and poses problems in the treatment of animals and humans. In recent years, resistance to beta-lactams has especially increased [10,11]. Although bacteria such as *Enterobacteriaceae* are reported to be the main source of beta-lactamases, they are also frequently found in Gram-positive bacteria. This resistance is acquired by exogenous genes or chromosomal mutations. However, bacteria can spread these genes horizontally and vertically among species [12,13]. For this reason, exposure to antibiotics is not a necessary condition for the development of resistance. Also, the transfer and spread of genes providing antimicrobial resistance are not only associated with environmental pathogens, but bacterial natural ecosystems are also determinant in this respect [14]. However, it was shown in previous studies that cow’s milk plays a role in the spread of methicillin-resistant *S. aureus* (MRSA), which is an extremely clinically important strain [15]. For this reason, mobile genetic elements can be faced in many different bacterial species. However, although the genes transferred to probiotic or commensal microorganisms and their resistance to antibiotics appear to be a positive condition, specific antibiotic resistance markers carried on mobile genetic elements pose a safety issue for health [16]. It is considered that overcoming this safety problem is possible with the “One Health Approach”, which focuses on the fact that infectious diseases and antimicrobial agent resistance are interconnected in terms of animal and human healthcare and must be continuously monitored together [17]. This study is the first report for Düzce, Türkiye. Thus, the first data about the state of the region are provided. Healthy-looking animals were selected as subjects. In the present study, the purpose was to identify staphylococcal species obtained from raw milk and to investigate some virulence characteristics.

## 2. Results

### 2.1. MALDI-TOF MS and Sequence Analysis

A total of 84 *Staphylococcus*-suspected isolates were obtained from 117 milk samples that were collected in the study. After some biochemical tests, the *Staphylococcus* spp. species were determined with MALDI-TOF MS in the isolates. The percentage dispersion of the isolates according to species is given in Figure 1. No weak spectrum was detected in the analysis and all of the isolates (100%) could be identified based on species. The score values of the isolates were determined between 1743 and 2786, which was found above 2000 for 79% of the isolates (Figure 2). However, no finding was detected with a score below 1700 (the cut-off value of the MALDI-TOF MS). The isolates were sent for post-PCR sequencing and 16rRNA characterization, and the results were found to be the same as for MALDI-TOF MS. In this respect, 84 bacteria including 10 different *Staphylococcus* species were obtained and the most isolated species was *S. aureus* (32.14%), followed by *S. borealis* (14.28%), *S. chromogenes* (10.71%), *S. warneri* (10.71%), and *S. epidermidis* (10.71%). *S. haemolyticus*, *S. hominis*, *S. xylosus*, and *S. vitulinus* species were detected at a rate of at least 3.57%.

### 2.2. Evaluation of Hemolysis, Biofilm, and Coagulase Ability

Although β-hemolysis was detected in 32.14% of the total isolates, α-hemolysis was not detected. The remaining isolates were evaluated as non-hemolytic. Although coagulase positivity was detected in 66.6% of *S. aureus* (*n* = 27) isolates in the species, strong biofilm formation was detected in 55.5% and hemolysis in 22.2%. Coagulase positivity was detected in some strains of *S. borealis*, *S. chromogenes*, *S. epidermidis*, and *S. hyicus*, and hemolysis was detected in some strains of *S. vitulinus*, *S. warneri*, *S. hyicus*, *S. chromogenes*, and *S. borealis*. In general, coagulase was detected in 46.42% of the total isolates (*n* = 84). Strong biofilm formation was detected in 57.14% of the *Staphylococcus* spp. isolates and 14.28% of them formed weak biofilms. The results determined by the slime-forming ability of the isolates on Congo red agar (Figure 3) showed parallelism with the biofilm formation that was determined by the microplate method. Some characteristics of the isolates according to the species are given in Table 1.

### 2.3. Antibiotic Resistance Profiles

The antibiotic susceptibility of the isolates was determined by bacterial growth inhibition zones (Figure 4). The antibiotic resistance of the isolates was observed at a relatively low level in the study. The highest resistance was found for kanamycin 21.42% and penicillin 19.04%, respectively. The least resistance was detected against 1.19% vancomycin and 3.57% ciprofloxacin. Two of the isolates were found resistant to at least four different classes, and 15 isolates were found resistant to least three different classes. The highest resistance was observed against the aminoglycoside class 40.54%. In addition, resistance to antibiotics in the beta-lactam group was observed in 39.63%. Multidrug resistance was observed in 20.23% of the isolates. The resistance profiles of the isolates are given in Table 1.

### 2.4. Beta-Lactamase Screening

The beta-lactamase presence of all the isolates were tested by chromogenic disk assay and positivity was detected in only seven isolates, four of which were *S. aureus* and three were *S. epidermidis* isolates. The color change in the positive isolates was from yellow to pink (Figure 5).

### 2.5. PCR Results

The presence of the *icaB* gene (57.14%) was detected in all strains of *S. aureus*, *S. borealis*, *S. chromogenes*, *S. hyicus*, *S. warneri*, and *S. epidermidis*, which formed strong biofilms by Congo red agar assay. No *ica* gene was detected in any of the *S. warneri*, *S. xylosus*, or *S. haemolyticus* strains, which were found to form weak biofilms in the microplate assay. The coagulase gene *coa* was was detected in 46.42% of all the isolates. The total ratio in the species was as follows: *S. aureus* (21.42%), *S. borealis*, *S. hyicus*, and *S. chromogenes* (7.14%), *S. epidermidis* (3.57%). The resistance gene *mecA* was detected in three of the *S. borealis* isolates, and both the *mecA* and *blaZ* genes were detected in three *S. epidermidis* isolates. Although seven isolates were found to be *mecA*-positive in the *S. aureus* strains, four *blaZ*-positive isolates were detected, including the *blaZ* gene in three *mecA*-positive isolates and the *blaZ* gene in one *mecA-negative* isolate. Also, no other strains that carried *mecA* and *blaZ* genes were detected, which means that 15.47% of 84 staphylococci were *mecA*-positive and 8.33% *blaZ*-positive. The virulence factors and PCR results of the isolates according to the species are given in Table 2 and Figure 6. The percentage evaluation of the virulence genes in each species is given in Figure 7.

### 2.6. Statistical Analysis

It was concluded that there is a complete correlation between biofilm forming and slime forming in all bacteria. Also, the data show that there is a strong correlation between biofilm-forming, slime forming, and the *icaB* gene. When all the virulence genes were evaluated collectively, a correlation was found between them. The statistical data are given in Table 3.

## 3. Discussion

Bovine milk is an important food source for humans and calves with its nutritive contents and probiotic microorganisms. However, it can also be a source of transmission for some foodborne diseases [18,19]. When first obtained, milk always has a microbial flora whose importance in terms of health is determined by the diversity of species and some characteristics of the species it includes [20,21]. Because of this, the purpose was to investigate and characterize the staphylococcal community in raw milk in the present study. There was no previous study in which flora was screened and the virulence characteristics were investigated in raw cow milk in the region where the milk samples were collected (Düzce, Türkiye). The only previous study was the screening of *Mycobacterium bovis* in raw milk samples [22]. Hereby, the present study is the first in terms of microbial screening in dairy cattle in this area. On the other hand, it can be considered that the number of samples collected in the study was limited. However, the results are considered important and will contribute to future studies because this is the first survey in the area. Microbial diversity and resistance development in bacteria must be constantly monitored in the environment–human–animal triangle, in both the wide and the narrow environments [23,24]. Especially considering that mastitis is the most important factor that causes economic losses in dairy cattle breeding in Türkiye, it would be useful to follow up on every farm and village-type farming in every area. Also, the identification of bacteria in milk shows the potential for subclinical mastitis [25,26,27]. The data obtained in this study are the first for the city of Düzce in Türkiye.

In general, the population of staphylococci is similar to that found in raw milk collected from healthy cows [3]. The identification findings in our study were similar to previous studies. The findings of the sequence analysis attest to MALDI-TOF MS identification. Therefore, it can be considered that MALDI-TOF MS is a highly accurate method for bacterial identification. MALDI-TOF MS has been used in microbiology practice for over 10 years and has almost replaced biochemical tests in terms of accuracy and costs, especially in the last 5 years. It is also an FDA-approved (Food and Drug Administration) method for microbial identification [28,29].

Some virulence factors determine the pathogenic importance of staphylococci opportunistic microorganism, which are persistent in the normal flora of milk [30]. Coagulase is one of the most important virulence factors for staphylococci. The *coa* gene was observed in two different sizes (approximately 850 bp and 550 bp) in this study. Because the *coa* gene encodes the coagulase protein, is highly polymorphic because of the variable sequences. With the repetitive sequence numbers, PCR products may have different lengths, even within the same species [31]. Javid et al. [31] have identified two different sizes of 595 bp and 802 bp for *coa* in their study, and these are so similar to our study. Six different genotypes of *coa*, 440 bp, 510 bp, 547 bp, 680 bp, 740 bp, and 820 bp, were found in another study [32]. According to another assessment, not only does *S. aureus* have a CoP characteristic, but other staphylococci also have a coagulase characteristic. There are studies in which *coa*-positive was detected even in strains known as CoN among *Staphylococcus* species. Unfortunately, other coagulase-positive staphylococci are ignored in veterinary medicine [33,34]. In this study, the *coa* gene was also detected in some of the *S. chromogenes*, *S. borealis*, and *S. epidermidis* strains, which are generally known as CoN. In the study of Javid [31], the *coa* gene was detected in 25% of *S. aureus* that was isolated from nasal swabs in cattle and in 86.3% of *S. aureus* isolated from milk with mastitis. In another study, it was reported that *coa*-positive and negative *S. hyicus*, *S. haemolyticus*, *S. epidermidis*, *S. chromogenes*, and *S. aureus* isolates were obtained in strains isolated from goat and sheep milk [34]. Eventually, the prevalence of CoPS (coagulase-positive staphylococci) was considered too high in this study because CoPS is usually pathogenic [32]. This is also an indicator of the genetic exchange between bacteria in natural environments. Some studies emphasize that commensal *Staphylococcus* spp. species is a good reservoir for virulence genes [35,36,37]. Gene transfer experiments were not performed in this study; however, they may be needed future studies.

Another virulence factor in staphylococci is biofilm formation. Biofilm-producing bacteria adhere to living/inanimate surfaces with the polymer structures they produce and imprison themselves in a matrix [38]. The presence of biofilm-forming strains in milk is clinically important because biofilm formation facilitates attachment to mammalian epithelial cells. Biofilms can protect bacteria from antibiotic stress and phagocytosis [3]. One study reported that 53.6% of *S. epidermidis* isolates were able to produce a biofilm [31]. Rudenko et al. [39] reported that high proportion of microorganisms isolated from cows with mastitis have the ability to form biofilms. Therefore, the presence of biofilm-forming strains in this study may be a potential hazard for mastitis formation. Researchers characterize biofilm as an important factor in terms of disease pathogenicity and drug resistance [40]. The *icaB* gene could not be detected in 12 strains that were found to form weak biofilms in phenotype tests. This may be because of other related genes. Previous studies show that *Staphylococcus* species can form biofilms independent of *ica* genes. The expression of other possible genes may be causing poor biofilm formation. There is no reason for the formation of a weak biofilm. Moreover, it may be a defect in the existing gene. Thus, there may also be deficiencies at the gene expression level. In general, biofilm formation takes place in two critical steps, attachment and accumulation. The adhesive proteins that provide this are encoded by many different genes. Additionally, in the transformative step, the bacterial cell surface province is complete, as it does not come into contact, but long-distance interactions occur between the bacterial cell and the surface. These are electrostatic forces, hydrophobic interactions, and van der Waals forces, which are weak interactions. Besides, the biofilm may not be passing through the maturation stage in these strains because biofilm formation is a multi-stage process [40,41,42,43,44]. Apart from this, another important virulence factor in staphylococci is the content of hemolysin. The strains damage the host cell membrane with their exotoxins, such as hemolysin, and transparent zones are formed in blood agar with the lysis of erythrocytes. Previous studies also reported that there are positive strains of hemolysis in staphylococci isolated from milk similar to our study [43,44,45,46].

Another issue that makes the bacteria in milk important for healthcare is the potential to carry antimicrobial resistance genes, because antibiotic consumption in animal husbandry can be unconscious in most developing countries [47,48]. Antibiotic resistance is not only a major crisis for non-healthy animals (with mastitis), but also a potential problem for healthy animals. Acute mastitis and chronic mastitis, which cannot be treated because of antibiotic resistance, cause loss of productivity in animals, as well as economic losses [6,49]. Hence, it is necessary to determine and monitor the resistance profile of bacteria in milk. In this study, the beta-lactamase content of staphylococci was investigated with the nitrocefin chromogenic test. Nitrocefin is a chromogenic cephalosporin, and the nitrocefin substrate is hydrolyzed by the β-lactamase producing culture upon reaction with the β-lactamase in bacteria, producing a colored compound. The nitrocefin test results were confirmed with PCR because the bacteria are genotype positive in some cases, although the chromogenic nitrocefin test is negative. For example, Bidya and Suman [50] investigated the phenotypic beta-lactamase with three different methods and reported that the highest accuracy belonged to the chromogenic nitrocefin test. In another study, the nitrocefin test yielded correct results for *S. aureus* and *S. epidermidis*, but was false-negative for *S. lugdunensis* [51]. In our study, both test results were compatible with each other.

Based on another perspective, the contents of the *mecA* and *blaZ* genes were investigated in the determination of the beta-lactam antibiotic resistance of the isolates in the present study. The *blaZ* encodes a protein (penicillinase) that inactivates penicillin by hydrolyzing the beta-lactam ring and appears to be the gene responsible for beta-lactam resistance in staphylococci [51]. The *mecA* encodes a penicillin-binding protein in methicillin-resistant *Staphylococcus* spp. For this reason, the presence of these two genes in bacteria is very important. However, the occurrence of these genes in natural ecosystems poses a threat to public and animal health [52]. The drug resistance profile of bacteria is constantly changing [12,13]. Different prevalences can be detected in studies conducted in different regions or in studies conducted in the same region at different times. In their study, Zhang et al. [53] detected 92.95% of the *blaZ* gene in *S. aureus* isolated from milk samples. In another study, 54% of the *blaZ* gene was detected in *Enterococcus* spp., which was isolated from sheep and goat milk, and attention was drawn to subclinical mastitis [54]. In this study, some isolates that contained both *blaZ* and *mecA* were evaluated as more critical because these two genes are very important virulence genes. The presence of the *mecA* gene is a common and growing crisis, especially in staphylococci. In his study, Abdeen reported that more than half of the isolates contained *mecA* [55]. Similarly, other studies are reporting that more than 50% *mecA* was detected in bacteria isolated from bovine milk [56,57]. The prevalence of *mecA* is a relatively low rate. However, as mentioned earlier, it poses a potential danger because resistance genes in bacteria are mostly mobile genetic elements and resistance development is in constant exchange among bacteria [58]. The multidrug resistance levels of the isolates in this study are moderately risky. On the other hand, some experts predict that there will be 10 million deaths each year associated with antibiotic resistance by the year 2050, and one of the major pathogens causing it will be *S. aureus* [59,60]. Based on a different aspect, more than 30% of *S. epidermidis* contained *mecA* and *blaZ* genes. This rate is relatively high when evaluated within the *S. epidermidis* population because the bacteria that contain *blaZ* are insensitive to beta-lactam antibiotics. The beta-lactam antibiotic group covers a large class and is a serious danger [61]. Also, although *mecA* is generally associated with methicillin resistance, studies show that *mecA* also facilitates resistance to other beta-lactam antibiotics [15,62]. Additionally, a correlation was observed between the presence of coagulase, *mecA*, and *blaZ* in biofilm-forming species. This shows that the strains contain more than one virulence factor.

The potential risks of antimicrobial agent resistance have focused on human health and concerns in the food sector. The fact is that antimicrobial agent resistance is a burden for many sectors throughout the world. Antimicrobial agent resistance screening should be performed in the field. The results of our study are the first for this region. However, the present situation must be monitored in the human–environment–animal triangle and resolved with the “One Health Approach”.

## 4. Materials and Methods

### 4.1. Sample Collection

The milk samples (117) were freshly collected directly from healthy animals (cattle) in 8 different villages from Düzce, Türkiye between March and May 2022 in 50 mL sterile falcon tubes. The animals were selected from village-type breeding barns in Türkiye, not from industrial dairy farms. They were not under any special care and reinforcement. There was only routine maintenance and feeding conditions. The history of illness and antibiotics use dated back at least 2.5 or 3 months for each of them. The milk samples were taken directly from the udder by hand milking method. Before the milking process, the udder was first washed with normal water, then wiped with sterile water and 70% alcohol. Disposable gloves were used during milking. The samples were taken from individual animals separately and were inoculated directly on nutrient agar (NA) (Condalab, Madrid, Spain) supplemented with 10% defibrinated sheep blood. After 24 h of incubation at 37 °C, different colonies were selected and included in the study. Tryptic soy agar (TSA) (Merck, Darmstadt, Germany) and tryptic soy broth (TSB) (Condalab, Spain) media were used for culture.

### 4.2. Isolation of Bacteria and MALDI-TOF MS Identificaiton

Following the incubation on blood agar, the colonies were purified into TSA medium according to their characteristics, such as colony morphology and pigmentation. The following tests were performed on each colony: Gram-staining, catalase (3.0% (*w*/*v*) H_2_O_2_), oxidase (1.0% tetramethyl-phenylenediamine dihydrochloride) OF (oxidation-fermentation) test results, growth abilities in the MacConkey agar (MAC) (Condalab, Spain) medium and the medium that contained 6.5% NaCl were recorded, respectively. The identification of the isolates that were found to be Gram-positive was performed with the matrix-mediated laser desorption ionization time-of-flight mMass spectrometry (MALDI-TOF MS) method. The MALDI-TOF MS device (Bruker Microflex LT, Bremen, Germany) and Flex Control 3.0 software were used for the microbial biomass analysis. In line with the manufacturer’s instructions, the scores between 2000 and 3000 were evaluated as “possible species identification”, scores between 1700 and 1999 were considered as “probable genus identification”, and scores below 1699 were evaluated as “unreliable genus identification”.

### 4.3. Molecular Characterization

The DNA extraction was performed with the TE (10 mM Tris-HCl and 1mM EDTA, pH:8.0) boiling method in all isolates. Molecular confirmation of identification was made with 16S rRNA gene sequence analysis of the isolates F- ATT CTA GAG TTT GAT CAT GGC TCA and with PCR R-ATG GTA CCG TGT GAC GGG CGG TGT GTA primers [63]. PCR master mix (K0171 Thermo Scientific, USA) 10 pmol reverse and forward primers and RNAse DNAse free PCR water were used in the reaction. Thermocycle initial denaturation was performed at 95 °C for 2 min, 1 min at 94 °C for 35 cycles, 1 min at 50 °C, 2 min at 72 °C, and 5 min at 72 °C after the last cycle. The resulting PCR products were sent to the Macrogen Company (Wageningen, The Netherlands) for sequence analysis. The comparison of the results was made with the GenBank Database by using the BLAST program.

### 4.4. Hemolysis Ability of Isolates

The prepared fresh cultures were inoculated on blood agar and the zone diameters around the colonies were evaluated after 24 h of incubation at 37 °C. Bright zone beta hemolysis, green zones alpha hemolysis, and no zone were considered non-hemolytic.

### 4.5. Congo Red Agar Assay

Congo red agar medium was used to determine the slime forming of the isolates. The Congo red agar (CRA) was prepared by using 0.4 g Congo red, 18 g sucrose, and 500 mL brain heart infusion agar (BHI) (all chemicals were obtained from Merck). The agar plates containing the inoculum of the bacteria were incubated at 37 °C for 24 h and then overnight at room temperature. Red colonies were defined as non-biofilm-forming strains, dark-colored colonies were defined as forming weak biofilms, and black colony-forming strains were defined as strong biofilm-forming bacteria [64].

### 4.6. Biofilm Assay

The biofilm formation was confirmed by microplate assay. A hundred µL bacterial culture (10^6^ CFU/mL) in TSB supplemented with 1% sucrose (Sigma-Aldrich, St. Louis, MO, USA) was added into each well of a sterile 96-well flat-bottom microplate. The microplate was left for incubation at 37 °C for 24 h. After the incubation, the absorbance was measured at OD_630_ nm using a microplate ELISA reader (Biotek BT 800, Winooski, VT, USA). Then planktonic cells were removed and each well was washed three times with sterile phosphate-buffered saline (PBS) (Sigma-Aldrich). The adherent bacterial cells were fixed at 60 °C for 45 min. After this fixing period, the cells were stained with 100 µL of 0.1% crystal violet (Himedia) for 15 min at room temperature (RT). Then the contents of each well were removed and washed 3 times with PBS again. After the 96-well plate was left to dry at RT for 15 min, 100 µL of 95% ethanol was added into each well to dissolve the bacteria on the bottom. The absorbance was measured at OD_490_ nm. The absorbance values were placed in the formula B = A490/A630. The results were evaluated as non-biofilm-forming microorganisms (B < 0.1), weak (0.1 < B < 0.5), moderate (0.5 < B < 1), and strong biofilm-forming microorganisms (B ≥ 1) [65].

### 4.7. Phenotypic Presence of Coagulase Enzyme

Rabbit plasma with EDTA was used to determine the coagulase properties of *Staphylococcus* spp. For the tube coagulase test, 0.5 mL of plasma was placed in each tube. A few colonies were inoculated from the cultures that were prepared the day before in TSA and incubated at 37 °C for 24 h. At the end of the incubation period, clot formation was considered positive. *S. aureus* ATCC 25923 was used as the control.

### 4.8. Multiple Antibiotic Resistance

The antibiotic resistance profiles of all the isolates were determined by a disk diffusion test recommended by the Clinical and Laboratory Standards Institute (CLSI) [66]. In this case, 15 different antibiotic discs from 7 different classes (Bioanalyse, Türkiye) were used in the study. These were vancomycin (30 µg), penicillin-G (10U), streptomycin (10 µg), tetracycline (30 µg), kanamycin (30 µg), neomycin (30 µg), nitrofurantoin (300 µg), erythromycin (15 µg), imipenem (10 µg), gentamicin (10 µg), amoxicillin-clavulanic acid (30 µg), ampicillin-sulbactam (20 µg), ciprofloxacin (5 µg), cefoxitin (30 µg), and oxacillin (1 µg). The test results were evaluated as sensitive (S), moderately sensitive (I), and resistant^®^. The *S. aureus* ATCC 25923 strain was used as a control. Multiple antibiotic resistance was considered as resistance to at least three different antimicrobial classes [67].

### 4.9. Chromogenic Nitrocefin Disk Method

The disk was wetted with 10 µL of distilled water. Suspicious colonies were applied directly on the disk in line with the manufacturer’s instructions and a color change (from light yellow to pink) within 15 min was evaluated as positive. A nitrocefin disk (Bioanalyse, Ankara, Türkiye) was used in the study.

### 4.10. Determination of coa, blaZ, mecA, and icaB genes

The primers used for *coa* (for the genotypic presence of coagulase enzyme), *icaB* (intercellular adhesion gene), and *blaZ* and *mecA* (resistance genes) in the isolates are given in Table 4. Sterile PCR water, PCR master mix, and 10 pmol reverse and forward primers were used for PCR mix. Each PCR reaction was run as 35 cycles with a total volume of 25 µL. In the reaction, the steps were pre-denaturation at 95 °C for 5 min, denaturation at 95 °C for 30 s, denaturation at 47 °C for *coa*, *blaZ* at 50 °C, *mecA* at 46 °C, *icaB* at 52 °C for 30-s primary bonding temperature, 30-s synthesis at 72 °C, and finally at 72 °C 5 min as the final synthesis. *S. aureus* ATCC 25923, *S. aureus* ATCC 29213, and *S. epidermidis* ATCC 35,984 were used as the positive control. The negative control was conducted with the PCR master mix and PCR water.

### 4.11. Statistical Analysis

The data from this study have been analysed by SPSS 17.0 software. The correlation between the biofilm and slime abilities of bacteria and the *icaB*, *coa*, *mecA*, and *blaZ* content was determined by a Pearson correlation test.

## 5. Conclusions

In conclusion, staphylococcal species were characterized in this study and *S. aureus* is the most isolated bacteria from milk samples. Furthermore, it was observed that all staphylococci contain various virulence factors. When evaluating the characteristics of all the bacteria, it can be seen that they have a high amount of coagulase enzyme and biofilm forming ability. Moreover, there are a lot of bacteria that contain a high level of other virulence factors, while the presence of antibiotic resistance genes remains relatively low in bacteria because the multidrug resistance rates are around 20% and the presence of *mecA* and *blaZ* is not very high. It may be useful to determine the prevalence of other genes responsible for resistance with future studies. Likewise, it may be more useful to increase the number of samples. This study is a preliminary screening because there have been no previous studies in the study region. These results do not only benefit the formation of a strategic plan in terms of subclinical mastitis and animal welfare, they may encourage further research. Knowing the presence of potential factors for mastitis can be useful for taking precautions. In addition, it must not be forgotten that humans and animals are in constant contact. Antibiotic resistance and health risk management caused by pathogens are only possible with a holistic health approach. In this respect, it is considered that the data on animal origin are very important because they constitute the data for the whole public health. Addedly, it should be remembered that antibiotic resistance is in a constant state of change between bacteria. Therefore, further studies are planned.

## Figures and Tables

**Figure 1 antibiotics-12-00556-f001:**
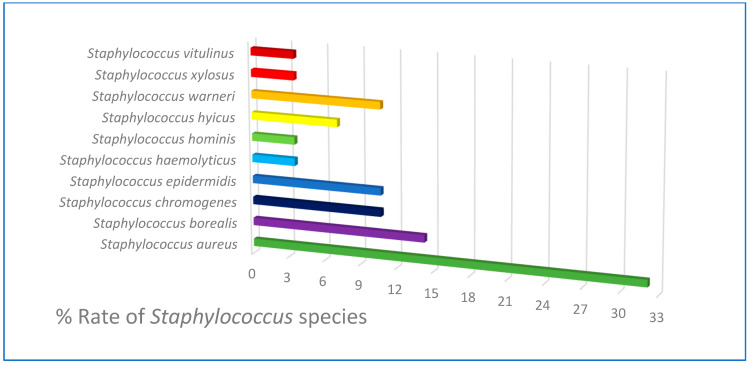
*Staphylococcus* species isolated from raw milk.

**Figure 2 antibiotics-12-00556-f002:**
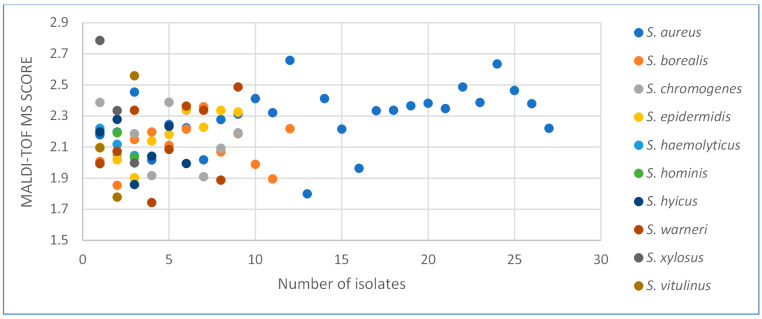
Dispersion of MALDI-TOF MS Scores.

**Figure 3 antibiotics-12-00556-f003:**
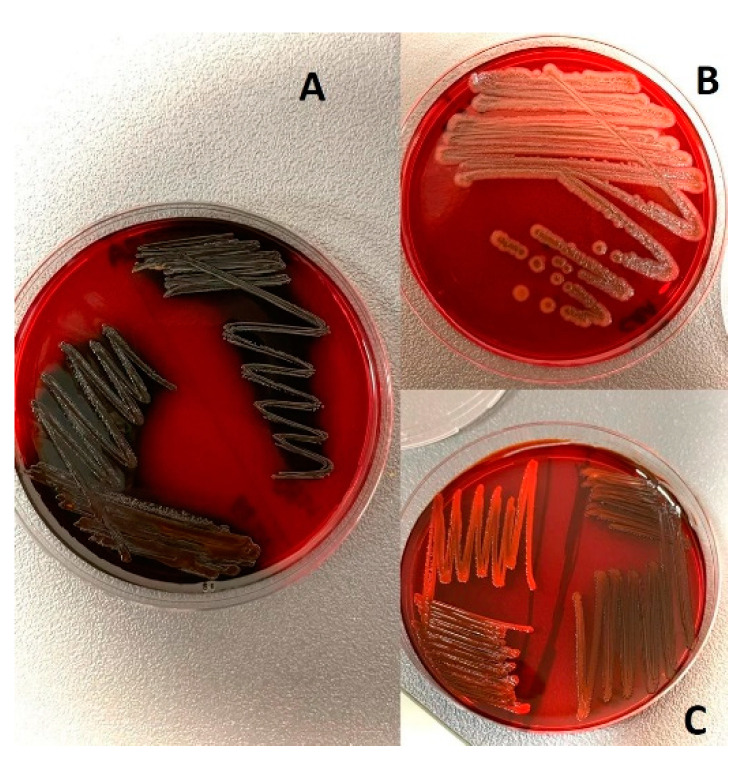
*Staphylococcus* spp. on CRA: (**A**) Strong slime-forming strain. (**B**) Non-producing slime. (**C**) Weak slime-forming strains.

**Figure 4 antibiotics-12-00556-f004:**
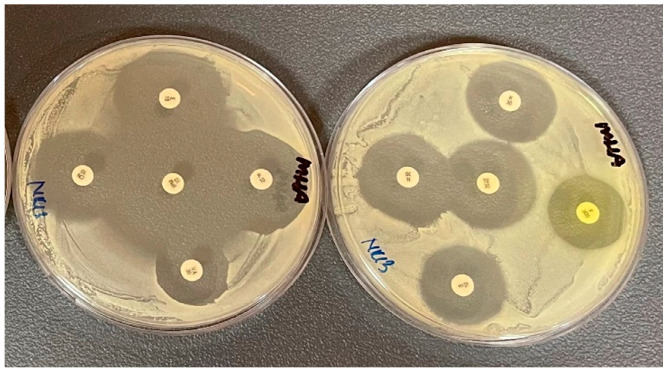
The images of disk diffusion assay on *S. aureus*.

**Figure 5 antibiotics-12-00556-f005:**
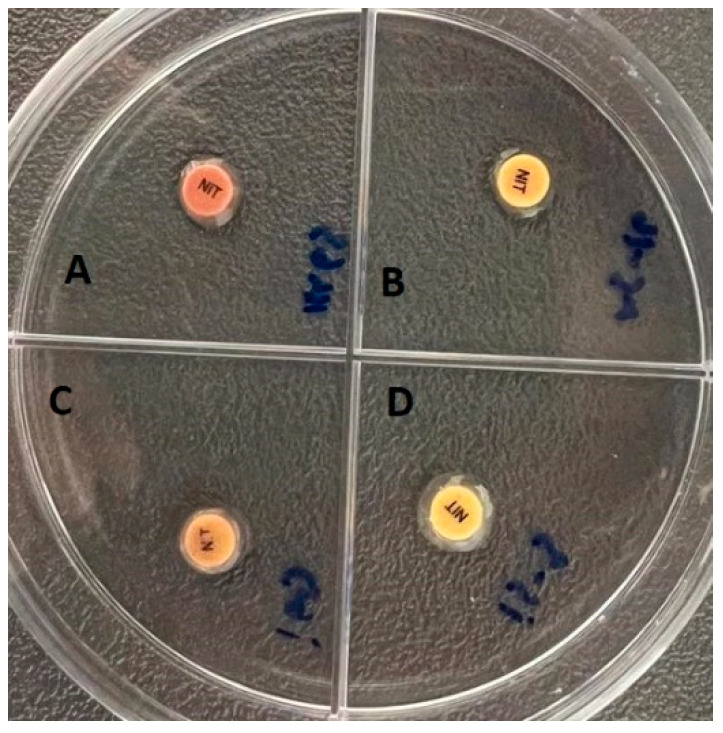
Chromogenic disk assay: (**A**,**C**) Beta-lactamase positive strains. (**B**,**D**) Beta-lactamase negative strains.

**Figure 6 antibiotics-12-00556-f006:**
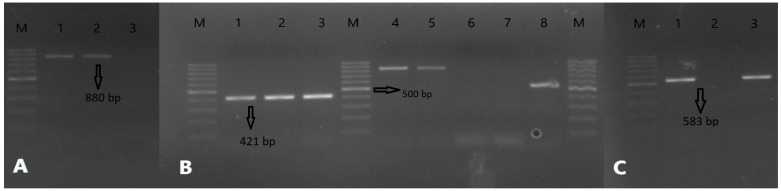
PCR results: (**A**) PCR bands for *icaB* (880 bp), 1,2: Positive sample, 3: Negative sample. (**B**) PCR bands for *blaZ* (421 bp) and *coa* (variable), 1: Positive control for *blaZ*, 2,3: Positive samples for *blaZ*, which is from the *S. aureus* strains, 4,5: Positive result for *coa* is almost 850 bp, 6,7: Negative samples, 8: Positive result for *coa* is almost 550 bp. (**C**) PCR bands for *mecA* (583 bp), 1,3: Positive samples for *mecA*, which is from the *S. epidermidis* strains, 2: Negative sample. M: Marker, 100–1000 bp DNA ladder is used.

**Figure 7 antibiotics-12-00556-f007:**
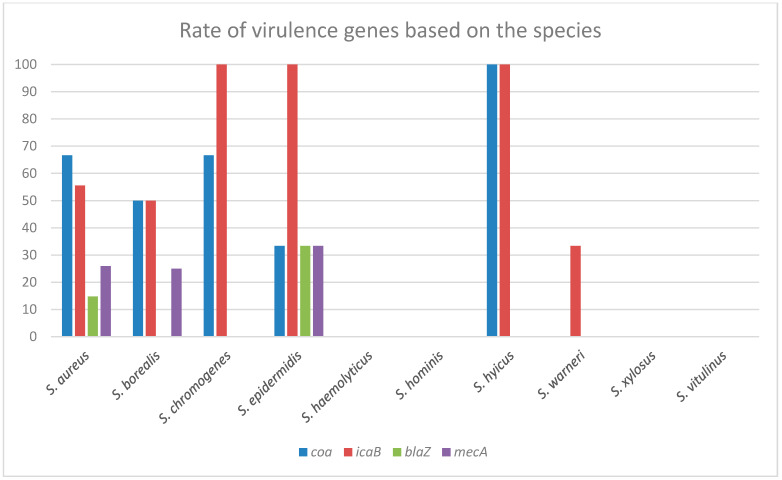
Distribution of virulence genes within the species: This table does not give the general distribution of virulence genes. In this table, information is given about how many of a species detected in the study have which gene. For example, while all *S. epidermidis* strains were *icaB* positive, more than 30% of them were positive for *coa, mecA*, and *blaZ*. At least six different species in the study contain one or more virulence genes. In the remaining four species, the virulence genes investigated in this study could not be detected.

**Table 1 antibiotics-12-00556-t001:** Number of antibiotic-resistant isolates.

Antibiotic Resistance (*n* = 84)					
Drug	R (*n*)	I (*n*)	Drug	R (*n*)	I (*n*)
TE	3	6	AMC	-	-
N	9	6	OX	15	-
K	18	6	FOX	13	-
CN	3	15	CIP	3	3
S	15	30	IMP	-	-
VA	1	-	E	15	36
F	-	12	P	16	-
SAM	-	-			

TE: Tetracycline, VA: Vancomycin, N: Neomycin, K: Kanamycin, CN: Gentamicin, S: Streptomycin, F: Nitrofurantoin, SAM: Ampicillin-sulbactam, AMC: Amoxicillin-clavulanic acid, IMP: Imipenem, E: Erythromycin, CIP: Ciprofloxacin, FOX: Cefoxitin, OX: Oxacillin, R: Resistant, I: Moderately sensitive.

**Table 2 antibiotics-12-00556-t002:** Number of isolates with positive test results.

Tür (*n* = 84)	Hemolysis	Weak Biofilm	Strong Biofilm	Coagulase	*coa*	*icaB*	*mecA*	*blaZ*
*S. aureus* (*n* = 27)	6	-	15	18	18	15	7	4
*S. borealis* (*n* = 12)	6	-	6	6	6	6	3	-
*S. chromogenes* (*n* = 9)	6	-	9	6	6	9	-	-
*S. epidermidis* (*n* = 9)	-	-	9	3	3	9	3	3
*S. haemolyticus* (*n* = 3)	-	3	-	-	-	-	-	-
*S. hominis* (*n* = 3)	-	-	-	-	-	-	-	-
*S. hyicus* (*n* = 6)	3		6	6	6	6	-	-
*S. warneri* (*n* = 9)	3	6	3	-	-	3	-	-
*S. xylosus* (*n* = 3)	-	3	-	-	-	-	-	-
*S. vitulinus* (*n* = 3)	3	-	-	-	-	-	-	-

**Table 3 antibiotics-12-00556-t003:** Correlations between virulence features.

Correlations							
		Biofilm	Slime	*icaB*	*coa*	*mecA*	*blaZ*
Biofilm	Pearson Correlation	1	1000 **	0.730 **	0.113	0.198	0.191
	*p*-Value		0.000	0.000	0.305	0.071	0.082
Slime	Pearson Correlation	1000 **	1	0.730 **	0.113	0.198	0.191
	*p*-Value	0.000		0.000	0.305	0.071	0.082
*icaB*	Pearson Correlation	0.730 **	0.730 **	1	0.372 **	0.304 **	0.261 *
	*p*-Value	0.000	0.000		0.000	0.005	0.016
*coa*	Pearson Correlation	0.113	0.113	0.372 **	1	0.196	0.151
	*p*-Value	0.305	0.305	0.000		0.075	0.170
*mecA*	Pearson Correlation	0.198	0.198	0.304 **	0.196	1	0.586 **
	*p*-Value	0.071	0.071	0.005	0.075		0.000
*blaZ*	Pearson Correlation	0.191	0.191	0.261 *	0.151	0.586 **	1
	*p*-Value	0.082	0.082	0.016	0.170	0.000	

* Correlation is significant at the 0.05 level (2-tailed). ** Correlation is significant at the 0.01 level (2-tailed).

**Table 4 antibiotics-12-00556-t004:** Primers used in the study.

Gene	5′–3′	Amplicon Size (bp)	References
*coa*	F-ACC ACA AGG TAC TGA ATC AAC G R-TGC TTT CGA TTG TTC GAT GC	500–1000	[55]
*blaZ*	F-CAAAGATGATATAGTTGCTTATTCTCC R-TGCTTGACCACTTTTATCAGC	421	[51]
*mecA*	F-AGA AGA TGG TAT GTG GAA GTT AG R-ATG TAT GTG CGA TTG TAT TGC	583	[55]
*icaB*	F-AGAATCGTGAAGTATAGAAAATT R-TCTAATCTTTTTCATGGAATCCGT	880	[43]

## Data Availability

All data in this study are presented in the submitted manuscript and in the figures.

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
