# Peer review of "The Characterization and Beta-Lactam Resistance of Staphylococcal Community Recovered from Raw Bovine Milk"

_antibiotics, 2023, doi:10.3390/antibiotics12030556_

Round 1
Reviewer 1 Report
This study characterize the Staphylococcus isolated from milk. Overall, the isolates have been well characterized, but they are needed to be evaluated at the phenotypic and genotypic levels. Specifically, no antibiotic resistance profiles of the isolates are provided. The antibiotic-associated genes were detected, which can be evaluated with the phenotypic properties such as MICs or disc diffusion results. In addition, biofilm-forming ability should be tested in this study. And, all gene names are written in italic font throughout the manuscript.
Minor comments
1. In some figures, the y-axis is not clearly marked.
2. Band sizes are added in PCR results in Figure 5.
Author Response
Thank you very much for your valuable suggestions. The shortcomings were largely attempted to be completed. We would very much like to have a better managed study and publish it together with your suggestions. Please do not hesitate to submit your new suggestions, if any. We hope that the publication has become convenient for you. Please find the revised manuscript.
We would like to point out that we did this study with limited facilities and that it is the first and only study in the region.
Kind regards.

Reviewer 2 Report
The paper The Characterization and Beta-Lactam Resistance of Staphylococcal Community Recovered From Raw Bovine Milk by Nisa Sipahi et al. described the occurrence of staphylococci in milk samples, selected characteristics related to their virulence and the presence of genes encoding beta-lactamases.
Due to the limited novelty, the article should not be published in its current form, additional research should be done and discussion and conclusions should be better written.
Whether milk samples were scratched from animals from large dairy farms or small farmers?
Whether the cows were milked by hand or using a mechanical milking system and whether hygiene standards were maintained?
Whether the animals had previous cases of mastitis or they were treated with antibiotics?
In addition to the beta-lactamase detection test, it is advisable to perform antimicrobial susceptibility test to determine the drug resistance profile of the tested strains.
Since the type of haemolysis was determined on a blood agar, it is also reasonable to detect genes encoding haemolysins hla and hlb using PCR method, as described , for example, in the publication Booth et al. 2001 [Booth, M. C., Pence, L. M., Mahasresthi, P., Callegan, M. C., Gilmore, M. S. 2001. Clonal Association among Staphylococcus aureus Isolates from Various Sites of Infection. Infect Immun., 69, 345-352].
The discussion is too long and repeats previously described results. Lack of a clearly set goal and conclusions. Does the presence of Staphylococcus in milk predispose animals to mastitis or can it pose a risk to consumers of unpasteurized milk?
A detailed review of the manuscript is hampered by the lack of line numbering
Table 2
Values ​​placed in it are badly formatted, e.g. ,000 does it mean 0 ; 1,000 or 1
Please standardize the writing throughout the text
1. Gram-negative or gram-negative , by analogy Gram-positive, gram-staining
2. Genes names should be italicized (icaB, mecA, blaZ)
3. staphylococci should be not italicized
4. 50ºC or 50°C e.g. p. 11 4.9. Determination of coa, blaZ, mecA, and icaB genes
5. Nitrocefin Cormogenic Test. Chromogenic
Page 8
The resistance of bacteria from cattle with mastitis or non-mastitis to important antibiotics is a current problem.
The sentence should be corrected as it is unclear
……falsely negative…………. correct to false negative
p. 10 Material Methods section
4.1. Sample Collection
More detailed explanation is required;:were the samples taken from individual animals or from milk mixed from different cows, right after milking. Are the selective media applied in the standard procedure for microbiological testing of raw milk. Were the animals were previously treated with antibiotics.
were inoculated on blood agar (Nutrient agar) that contained 10% defibrinated sheep blood, prepared using Condalab) …. or blood agar – nutrient agar supplemented with 5% sheep blood
blood agar is usually prepared from Tryptic Soy Agar or Columbia agar base with 5% sheep blood
4.2. Isolation of Bacteria and MALDI TOF MS
H2O2 or H2O2
MAC medium - what is the substrate MAC
4.8. Chromogenic Disk Assay
The disk was wetted with 10 μl of distilled - wrong font size
p. 12 5. Conclusions
In conclusion, it was found that Staphylococcus spp. isolated from raw milk in this study contained some virulence factors at a high level, but contained a relatively low level of antibiotic resistance genes.
Level or rather the frequency of occurrence ???
Author Response

(The authors gave the same response as above.)

Reviewer 3 Report
1. The specific reason for the formation of weak biofilm is uncertain whether it is affected by other genes or gene defects.
2. In 4.9 gene detection, the negative control should be a system without primer, not distilled water.
3. The agar plate mentioned in 4.5 can be stored overnight in a 37 ℃ incubator.
4. The resistance gene was screened, but no drug sensitivity test was done.
5. The sample collection should have been taken from several different areas, the sample size was too small.
6. The sample size of Staphylococcus aureus could be compared to the number of cows being affected by mastitis to see how much pathogenicity there is.
7. Three-wire table is recommended for Table 2
8. Table 2 has different formats, and the content is suggested to be centered
9. The number of reference authors should be limited to three
10. Increase the sample size a little bit
11. The "blaZ" in the article is the gene and should be italicized
12. In the PCR result, whether the marker to the right of lane 8 can be used, Whether the size should be marked on the marker?
13. The author of the reference keeps the first three.
14. Whether the culture has been grown on the blood AGAR for too short a time.
15. The cause of resistance gene transmission was not specifically investigated.
Author Response

(The authors gave the same response as above.)

Round 2
Reviewer 1 Report
This is well revised based on the comments.
Author Response
There is no any request for revision. I would like to thank to Reviewers for their contributions. I tried to do my best.
I hope that Editor thinks the same. Minor revisions related to English Grammar has been done.
Kind Regards.
Reviewer 2 Report
The manuscript was revised according to the guidelines included in the previous review. Unfortunately, the current version also has shortcomings.
Please pay attention to the conclusions, which should be short and reflect the results obtained by the authors.
Detailed comments
L 100 S. aureus italic
L 104 Staphlococcus …. Staphylococcus
L 106 congo red agar Congo Red Agar
L 98 and 108 Antibiotic Resistance Profile …. please move the phrase to the next line
2.3. Antibiotic Resistance Profile
L109 inhibition zones … change to bacterial growth inhibition zones
L 125 The images of disk diffusion assay on Bacteria ….. please specify what strain is in the photo
L 138 The icaB … The presence of the icaB gene
L 139 No icaB No ica gene
L 141 coa was detected ….Coagulase gene coa was …
L 145 mecA-negative .. mecA-negative
Table 3 ,730** ,372** please change to 0.730 0.372
From page 6 no continuation of verse numbering; on page 7 verse numbering started from 1.
P 10 L 108 The resistance constantly change in bacteria [12, 13] … The drug resistance profile of bacteria is constantly changing.
P 12 L 213 4.9. Chromogenic Nitrocefin Disk Method …. please move the phrase to the next line L214
P 13 L 232 Conclusions
The conclusions are too general they do not describe the results obtained but commonly known facts. Please summarize to the most important conclusions.
Author Response
Dear Reviewer
I would like to thank you for your contributions. Manuscript has been revised.
Kind Regards.
